# Active Surveillance of Powassan Virus in Massachusetts *Ixodes scapularis* Ticks, Comparing Detection Using a New Triplex Real-Time PCR Assay with a Luminex Vector-Borne Panel

**DOI:** 10.3390/v16020250

**Published:** 2024-02-04

**Authors:** Guang Xu, Eric Siegel, Nolan Fernandez, Emily Bechtold, Timothy Daly, Alan P. Dupuis, Alexander Ciota, Stephen M. Rich

**Affiliations:** 1Department of Microbiology, University of Massachusetts, Amherst, MA 01003, USA; gxu@umass.edu (G.X.); nolansmithfernandez@gmail.com (N.F.); emily.bechtold@colostate.edu (E.B.); tdaly123@gmail.com (T.D.); 2New York State Department of Health, Wadsworth Center, Slingerlands, NY 12159, USA; alan.dupuis@health.ny.gov (A.P.D.II); alexander.ciota@health.ny.gov (A.C.)

**Keywords:** Powassan virus, *Ixodes scapularis*, real-time PCR, Luminex, Massachusetts

## Abstract

Powassan virus is an emerging tick-borne pathogen capable of causing severe neuroinvasive disease. As the incidence of human Powassan virus grows both in magnitude and geographical range, the development of sensitive detection methods for diagnostics and surveillance is critical. In this study, a Taqman-based triplex real-time PCR assay was developed for the simultaneous and quantitative detection of Powassan virus and Powassan virus lineage II (deer tick virus) in *Ixodes scapularis* ticks. An exon–exon junction internal control was built-in to allow for accurate detection of RNA quality and the failure of RNA extraction. The newly developed assay was also applied to survey deer tick virus in tick populations at 13 sites on Cape Cod and Martha’s Vineyard Island in Massachusetts. The assay’s performance was compared with the Luminex xMAP MultiFLEX Vector-borne Panel 2. The results suggested that the real-time PCR method was more sensitive. Powassan virus infection rates among ticks collected from these highly endemic tick areas ranged from 0.0 to 10.4%, highlighting the fine-scale geographic variations in deer tick virus presence in this region. Looking forward, our PCR assay could be adopted in other Powassan virus surveillance systems.

## 1. Introduction

Powassan virus (POWV) is a tick-borne flavivirus that can cause fatal encephalitis in North America [1,2]. The virus was first isolated from neural tissue in a child who had died in the 1950s of encephalitis in the town of Powassan, Ontario, Canada [3]. In the United States, the first human case of POWV disease was reported in 1970, and neuroinvasive disease was formally added to the list of nationally notifiable diseases in 2001 [4,5]. Reported human cases have increased drastically to a record 290 reported cases in 2022 from only 1 case each year between 2004 and 2006, mostly from 12 states in the northeast and north-central United States [5]. More than 10% of the reported cases resulted in death, and 50% in long-term neurological damage [5,6].

Powassan virus has been found in four *Ixodes* tick species (*I. cookei*, *I. marxi*, *I. scapularis* and *I. spinipalpus*) and one *Dermacentor* tick species (*D. andersoni*) through active tick surveillance [7,8,9,10]. Sequence analysis shows that POWV is comprised of two genetic lineages, including the prototype lineage (POWV, lineage I) and a second lineage that was first detected in *Ixodes scapularis* ticks (POWV lineage II or Deer Tick Virus, DTV) [11]. Although these two lineages are serologically and clinically indistinguishable, they have distinct enzootic cycles and differ in the threat they pose to public health [2,12]. In nature, prototype POWV is maintained predominantly by *I. cookei* and *I. marxi* ticks and their respective mammal hosts, groundhogs (*Marmota monax*) and squirrels [7,13]. The main enzootic cycle for DTV maintenance has long been thought to be between *I. scapularis* and the white-footed mouse (*Peromyscus leucopus*) [14]. Shrews (*Sorex* spp.; *Blarina brevicauda*) have recently been implicated as reservoir hosts for DTV, and many unknowns still exist around DTV ecological maintenance and amplification [15,16]. Deer Tick Virus is considered a greater public health threat due to the opportunistic nature of *I. scapularis* [17,18]. *Ixodes scapularis* is the most frequent human-biting tick in the United States and is responsible for the transmission of several other human disease-causing agents including those responsible for Lyme disease, anaplasmosis, and babesiosis [19].

The experimental demonstration of POWV transmission within 15 min of tick attachment necessitates fast and easy detection [20]. Several methods have been used to identify POWV in ticks [21]. Virus culture and antigenic testing were the most common methods employed prior to the advent of polymerase chain reaction (PCR) methodologies in the 1990s [7,10]. To date, serological diagnostics through detection of POWV-specific IgM antibodies with a confirmational plaque-reduction neutralization test (PRNT) remains the current standard for diagnosing infection in human patients [22]. Since RNA cannot be amplified directly using a PCR, a reverse-transcriptase is required prior to PCR in both conventional and real-time formats to identify POWV RNA [11,14,18,23,24,25,26,27,28]. The quality of pathogen testing in human biting ticks is greatly increased when internal control amplifications targeting tick genes are done in conjunction with the pathogen test(s) [27,28]. For testing viral pathogens in ticks, an internal control is even more desirable since RNA molecules tend to be less stable, thus increasing the possibility of false negative results.

A new diagnostic tool uses the Luminex xMap technology, a bead-based array platform that can detect up to 100 different DNA targets simultaneously [29]. The Luminex xMAP MultiFLEX Vector-borne Panel 2 (Luminex panel) is a multiplex, nucleic-acid-based, commercially available kit for detection of multiple disease-causing agents simultaneously. Any single or combination of 12 different DNA and/or RNA targets, including POWV, can be selected to create a custom assay. The Luminex approach is dependent upon a conventional PCR amplification prior to detection in the bead-based assay. Sensitivity and specificity of the Luminex panel for POWV have not been compared with real-time PCR. The present study describes the development and validation of a quantitative real-time, multiplex PCR assay (with appropriate tick internal control) for the rapid sensitive and specific detection of POWV and DTV in *I. scapularis*. We compare this method with the Luminex panel in an active surveillance study of tick populations in the coastal mainland and islands of Massachusetts, focusing on 13 sites from the highly endemic regions of tick-borne diseases in Cape Cod and Martha’s Vineyard Island.

## 2. Materials and Methods

### 2.1. Study Sites and Tick Collection

Ticks were collected from 6 sites on Cape Cod (Barnstable County) and 7 sites on Martha’s Vineyard Island, Massachusetts, from March to May 2016 (Figure 1). Mainland sites were an average of 20 km away from each other and covered 83 km^2^. Ticks were collected by dragging a 1 m^2^ white fabric over tick-infested vegetation. After collection, ticks were frozen individually at −80 °C until species identification and nucleic acid extraction.

### 2.2. Tick Species Identification and Total Nucleic Acid Extraction

Species identification of *I. scapularis* ticks was conducted using published identification keys, and confirmed by a species-specific TaqMan PCR assay [30,31]. Total nucleic acids were extracted from individual ticks using MasterPure Complete DNA and RNA Purification Kits (LGC Biosearch Technologies, Madison, WI, USA) following the manufacturer’s protocol. In brief, the tick was homogenized in lysis solution, digested with proteinase K, and incubated (65 °C for 15 min). The digested proteins were precipitated with MPC Protein Precipitation Reagent. Supernatants were transferred to clean microcentrifuge tubes where nucleic acids were precipitated with isopropanol and washed twice with ethanol. After drying, the pellets were resuspended in water (40 µL).

### 2.3. Taqman Real-Time PCR Assay

A new triplex TaqMan assay with an RNA internal control was designed and performed along with a published singleplex assay [18]. The TaqMan probe and primer sequences were designed to be specific for POWV and DTV based on ns5 genes (Table 1). To assess assay specificity, the primer and probe sequences were initially checked using the BLAST program [32]. The assay was then tested with genomic DNA from humans, mice (Promega, Madison, WI, USA), ticks (*Amblyomma americanum*, *D. variabilis* and *I. scapularis*) and common tick-borne pathogens present in the United States: *Borrelia burgdorferi*, *B. miyamotoi*, *B. lonestari*, *Anaplasma phagocytophilum*, *Rickettsia rickettsia*, *Francisella tularensis*, *Ehrlichia chaffeensis*, Colorado Tick Fever virus and Heartland virus.

An internal RNA control for *I. scapularis* ticks was designed with a Cy5 labeled probe. The amplicon spans an exon–exon junction of the *Ixodes calreticulin* gene to target only reverse-transcribed mRNA and avoid detection of tick-genomic DNA. This probe was tested with tick DNA, RNA and total nucleic acid to confirm it only detects the *Ixodes* tick RNA target. The HEX-labeled degenerate probe detects POWV lineage I and II, and the FAM-labeled probe specifically detects POWV lineage II (deer tick virus, DTV) (Table 1).

We performed Taqman real-time PCR assays in a triplex format with 16 μL reaction volumes using the Brilliant III Ultra-Fast QRT-PCR Master Mix (Agilent, La Jolla, CA, USA) in an Agilent MX3000P qPCR System. The primers and probes were optimized at concentrations of 100 nM, 200 nM, 300 nM and 400 nM. Cycling conditions included an initial cDNA step at 50 °C for 10 min, then an activation of the Taq DNA polymerase at 95 °C for 3 min, followed by 40 cycles of 15 s denaturation at 95 °C and 1 min annealing-extension at 60 °C. Water was run as a negative control for each test. The double-stranded, sequence-verified gene fragments, gBlock DNA fragments (Integrated DNA Technologies, Inc., Coralville, IA, USA) of the tick *calreticulin* gene and POWV ns5 gene were used as positive controls and for standard curve construction. 

### 2.4. Internal Control

In the triplex assay, a single-copy *calreticulin* gene, conserved in *I. scapularis*, was used as an internal control. The probe spans an exon–exon junction of *calreticulin* gene to avoid tick genomic DNA detection because the gene coding region has only one intron with a conserved position [33]. The internal control was tested with tick DNA, RNA and total nucleic acid, confirming its specificity for *I. scapularis* tick RNA, not DNA.

### 2.5. xMAP MultiFLEX Vector-Borne Panel

Ticks were also tested on the Luminex xMap magnetic bead-based technology following the manufacturer’s xMAP MultiFLEX Vector-borne Panel 2 protocol. Targets were first amplified under the following conditions: 12.5 µL Takara 2× Buffer, 1 µL Takara OneStep RT-PCR Enzyme, 5 µL Luminex Vector-borne Panel 2 Primer Mix, 3.5 µL RNase free water and 3 µL total nucleic acid extraction; initial 1 cycle at 50 °C for 30 min, 1 activation cycle at 95 °C for 15 min, 40 cycles of denaturation at 95 °C for 45 s with annealing and extension at 60 °C for 45 s.

Following the RT-PCR, the amplified product was hybridized with the magnetic bead solution and rinsed with SAPE for analysis. The bead solution was prepared by diluting 3 µL of Bead Mix into 7 µL 1× TE and 30 µL Buffer A for each reaction. For each sample, 40 µL of diluted bead mixture was mixed with 10 µL of PCR product and incubated at 95 °C for 4 min, followed by 52 °C for 15 min. Following hybridization, the plate was placed on a magnetic plate separator to allow the beads to separate and pellet for 2 min. After removing the supernatant, each sample was mixed with 75 µL of the diluted SAPE solution (0.3 µL stock SAPE and 74.7 µL Buffer B) and incubated at 52 °C for 15 min.

The sample plates were then analyzed on the Luminex MagPix system for the median fluorescence intensity by subtracting the MFI of the background, calculated using a negative control from the MFI of each sample according to the manufacturer’s Vector-Borne Panel 2 instructions. Data were analyzed using the xTAG Analysis Software LSM 2.4 per manufacturer’s recommendations.

### 2.6. Assay Comparison

To evaluate the performance of the triplex assay, we compared the results with those of a singleplex real-time RT-PCR assay (Wadsworth Center) and the Luminex panel for 819 field-collected ticks [18]. Primers FWD-CATAGCRAAGGTGAGATCCAA, REV-CTTTCGAGCTCCAYTTRTT and probe-AGCTCTGGGCGCATGGTYGGATGAACA were used in the Wadsworth Center singleplex assay for the detection of POWV. Another set of primers and probe set was used for confirmation of DTV isolates (FWD-GATCATGAGAGCGGTGAGTGACT, REV-GGATCTCACCTTTGCTATGAATTCA and Probe-TGAGCACCTTCACAGCCGAGCCAG) [18]. The sensitivity of the PCR tests was assessed against each other with a *t-*test of cycle threshold (Ct) values.

## 3. Results

### 3.1. Real-Time PCR Powassan Virus Testing of Field-Collected Ticks

Of 819 field samples, 33 ticks (4.03%) tested positive for POWV in both triplex assay and Wadsworth Center singleplex PCR assays. A total of 752 ticks were tested as POWV negative, and 34 ticks (4.15%) failed the internal tick RNA control in the triplex assay. POWV was detected in *I. scapularis* at four out of six sites in Cape Code, and two out of seven sites in Martha’s Vineyard, Massachusetts. Infection rates varied across sites, reaching as high as 10.43% at the Truro, Cape Cod site, and showing complete absence at seven sites (Table 2 and Table 3). All ticks that tested positive for POWV were positive for DTV, and no cases of lineage I infection were detected.

### 3.2. Comparison of Taqman Real-Time PCR Assay with xMAP MultiFLEX Panel

Both Wadsworth Center singleplex and our triplex real-time RT-PCR assays detected the 33 POWV positives, with an average Ct value 27.22 for the triplex assay and 28.81 for the Wadsworth Center assay. There was no statistical significance in Ct values between the two real-time PCR methods (*p* = 0.2739), suggesting that these two qPCR assays had similar sensitivity. However, the triplex real-time PCR assay detected 34 ticks (4.15%) that failed the internal RNA quality control, a feature not available in the Wadsworth Center assay.

Both real-time PCR assays appeared to demonstrate higher sensitivity than the Luminex panel. Out of 819 ticks, the Luminex panel identified 30 POWV-positive ticks, whereas both real-time PCR assays detected 33 (Table 3). The three POWV-negative ticks, as determined by the Luminex panel but positive as determined by real-time PCR, had Ct values greater than 31, which indicated a low POWV load in these samples. This suggested a lower limit of detection for the real-time PCR assay than Vector-borne Panel.

The specificity of the Luminex panel matched that of the real-time PCR testing. However, the Vector-borne Panel considered 34 ticks (4.15%) that failed the real-time PCR RNA quality control as POWV negatives, as it lacked a built-in tick RNA quality control.

## 4. Discussion

This study had three aims: (1) develop a reliable triplex real-time PCR method to detect the presence of POWV and distinguish POWV lineages in *Ixodes* ticks with an internal control; (2) compare this assay’s performance to the commercially available Luminex panel; and (3) apply these assays in active surveillance of POWV in *I. scapularis* populations in Massachusetts.

In our triplex assay, the primers and probes were evaluated for possible cross-reactivity with bacterial, parasitic, and viral tick-borne pathogens. These were done in silico by sequence alignment using BLAST, and experimentally by testing the assay against tick-borne DNA and RNA pathogens. No cross-reactivity was observed. The sensitivity of the new triplex assay is comparable to the Wadsworth Center singleplex assay, but is evidently higher than the Vector-borne Panel assay, as we saw that 33 of 819 ticks were POWV positive for both triplex and singleplex real-time assay, while only 30 were POWV positives when using the commercial Vector-borne Panel. These discordant results between real-time PCR and Vector-borne Panel assay were associated with low POWV loads in ticks. Confirmation of the difference in the sensitivity observed may be done in the future with confirmatory testing considering virus-containing tick suspensions diluted to different extents. Studying these samples using RT-PCR and subsequent Sanger sequencing of the PCR product would also be beneficial to rule out the possibility of false positive results. The triplex and Wadsworth Center singleplex assays can detect POWV I and lineage II, but the Vector-borne Panel can only detect general POWV.

Ideally, tick-borne pathogen detection includes the use of internal controls because an assay failure due to nucleic acid extraction, PCR, or PCR inhibition can generate a false negative result. False negative results may occur in field collected and blood-fed ticks because: (1) Sample preservation in the field to ensure RNA and DNA stability is not always possible. RNA degradation is noticeable within hours [34]. (2) Trace amounts of blood in ticks can inhibit PCR reactions [35]. Using established methods, instances of degraded RNA or blood-fed ticks have usually been treated as negative samples due to the absence of an RNA internal control.

The 16S ribosomal RNA gene and beta-actin gene are commonly used as RNA internal controls for ticks [36,37]. However, specific amplification is cumbersome because of genomic DNA (gDNA) contamination in tick RNA samples. In this new triplex assay, we used a probe that spans an exon–exon junction of *I. scapularis* tick *calreticulin* gene as an internal control to verify RNA quality (Table 1). This design reduces the risk of false positives from amplification of contaminating tick genomic DNA, since the intron-containing genomic DNA sequence would not be detected. It also reduces the incidence of false negatives by providing a clear indication of failed RNA extraction or PCR inhibition through the internal RNA quality control, enabling the accurate interpretation of negative results. In this study, 34 of 819 (4.15%) field-collected ticks failed the internal RNA quality control. Although a negative result was obtained for these RNA-quality-control-failed ticks in triplex and Vector-borne Panel assays, these could have been false negative results. The Luminex panel assay underestimated the POWV prevalence rate due to lack of internal tick RNA quality control. With its higher sensitivity and built-in internal quality control, the triplex assay is an improvement over current detection methods.

Our final aim was to apply these assays to active surveillance of POWV in field-collected ticks on Cape Cod and Martha’s Vineyard Island, where ticks and tick-borne *B. burgdorferi*, *A. phagocytophilum* and *B. microti* are endemic [38]. Powassan virus was detected in *I. scapularis* at 4 of 6 sites in Cape Cod and 2 of 7 sites in Martha’s Vineyard at infection prevalence between 0 and 10.43%, higher than previous findings in host-seeking ticks surveyed in surrounding states, including Connecticut (0–4.2%), New York (0–4.9%, 0–3.4%), Maine (0–3.5%), Wisconsin (0–1.3%) and Rhode Island (0–0.6%) [8,9,14,25,39]. In the present study, the POWV infection was 4.20% in Cape Cod and Martha’s Vineyard on average. Our results demonstrate that POWV infection rate in *I. scapularis* is variable among sites, but with a relatively high infection rate in Cape Code and Martha’s Vineyard Island.

The mechanisms behind the focality of DTV transmission cycles in Massachusetts and the surrounding northeast states remain poorly understood [16]. The variation in site-specific infection prevalence (0–10.4%) on a relatively small scale observed in our survey of host-seeking ticks may be partially attributed to the limited sampling power and scale inherent in active surveillance efforts. However, these differences could also reflect sporadic amplification of DTV in mammalian reservoirs such as shrews and others not yet identified, and possibly even co-feeding events, which could be facilitating differences in the presence of DTV in host-seeking *I. scapularis* across fine spatial scales [15]. Further research is needed to better understand the roles of reservoirs, co-feeding, transstadial and transovarial transmission, and other ecological factors driving the presence of DTV in tick populations.

We suspect that our results reflect increasing DTV infection rate in Massachusetts in recent years. In 1997, a 0.64% POWV infection rate was reported in northeastern Massachusetts, and in 2016 we found an average 4.20% infection rate in ticks from Cape Code and Martha’s Vineyard Island [11]. This aligns with a significant increase in seroprevalence of POWV in New England deer, where 91% were found seropositive for POWV in 2009, compared to only 4% in 1979; and this also aligns with the increased incidence of human POWV infection in Massachusetts and elsewhere in the United States [5,40]. The first case of encephalitis attributed to POWV in Massachusetts was reported in 1994 [41]. This remained the only case reported there until a single case followed in 2013. Since 2013, testing has become more widespread, and 49 cases have been reported there through 2022, accounting for 17% of reported cases in the United States since 2013 [5]. This highlights the significant public health threat that POWV poses to this region.

As expected, we found only DTV in *I. scapularis* in this study. Prototype POWV was not detected. Prototype POWV is considered to be a minor public health concern due to the relative host specificity of its tick vector, *I. cookei* and/or *I. marxi* [14]. The deer tick virus is well established in Massachusetts. It has proven to be a greater public health threat since its discovery in 1997 because it is associated with *I. scapularis* and is therefore more likely to come in contact with humans [11,17,18,41]. The deer tick virus may be a particularly significant public health threat in the east coast and islands of Massachusetts, where we surveyed in the present study, considering (1) the increasing human biting tick population in the area [38]; (2) a relatively higher POWV infection rate at some sites; and (3) a recent increase in POWV human cases in Massachusetts [42,43].

## 5. Conclusions

Our study describes a qPCR-based triplex assay to simultaneously detect POWV Lineage I and II with a tick exon–exon junction RNA internal control. We demonstrate an increase in sensitivity and quality control over standard methods, and hope that this technique will be useful in enzootic transmission studies and as a tool for the monitoring and prevention of POWV.

## Figures and Tables

**Figure 1 viruses-16-00250-f001:**
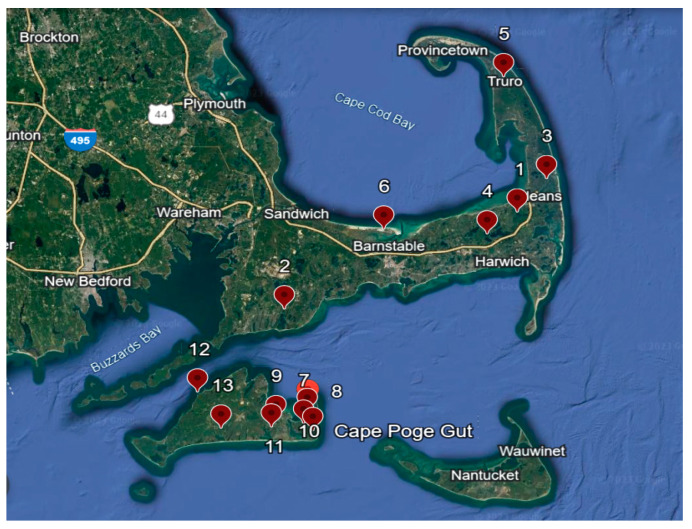
Tick collection sites along coastal mainland and islands of Massachusetts, USA.

**Table 1 viruses-16-00250-t001:** Taqman real-time triplex PCR assay to detect Powassan (POWV) virus (lineage I and II) and POWV lineage II (deer tick virus, DTV) with *Ixodes scapularis* RNA as an internal control.

Target	Gene	Type	Sequences (5′-3′)	Con. (nM)
*Ixodes* tick RNA	*calreticulin*	Forward	CCAAGGTGTACCTCAAGGAAGAG	200
		Reverse	TTGAAAGTTCCCTGCTCGCTTC	400
		Probe	Cy5-ZEN-TCGCCGACGGAG(INTRON)ACGCCTGGAC- Iowa Black RQ	300
		Standard Curve	Y = −3.469 × LOG(X) + 15.11, Eff. = 94.2%, RSq = 99.8%	
POWV lineage I and II	ns5	Forward	TGACAGACACAACAGCGTTTGG	200
		Reverse	TCACTCACHGCTCTCATGATCAC	300
		Probe	HEX-ZEN-CTGGTGCCTGGCTGYGGYTCYTGGG- Iowa Black FQ	100
		Standard Curve	Y = −3.578 × LOG(X) + 16.29, Eff. = 90.3%, RSq = 99.4%	
POWV lineage II (DTV)	ns5	Forward	GATCATGAGAGCGGTGAGTGACT	400
		Reverse	GGATCTCACCTTTGCTATGAATTCA	400
		Probe	FAM-ZEN-TGAGCACCTTCACAGCCGAGCCAG- Iowa Black FQ	300
		Standard Curve	Y = −3.541 × LOG(X ) + 18.46, Eff. = 91.60%, RSq = 99.7%	

**Table 2 viruses-16-00250-t002:** Prevalence of Powassan (POWV) virus and POWV lineage II (deer tick virus, DTV) in *Ixodes scapularis* ticks collected in six sites in Cape Cod and seven sites in Martha’s Vineyard, Massachusetts.

Location	Site	*n* Ticks Collected	*n* Ticks Passed RNA Control	POWV All+	DTV+
Cape Cod, Brewster	Nickerson State Park (1)	130	129	9 (6.98%)	9 (6.98%)
Cape Cod, East Falmouth	Waquoit (2)	127	101	5 (4.95%)	5 (4.95%)
Cape Cod, Eastham	Fort Hill (3)	118	118	3 (2.54%)	3 (2.54%)
Cape Cod, Harwich	Punkhorn (4)	116	114	0 (0%)	0 (0%)
Cape Cod, Truro	Truro (5)	116	115	12 (10.43%)	12 (10.43%)
Cape Cod, West Barnstable	Sandy Neck (6)	38	38	0 (0%)	0 (0%)
Martha’s Vineyard, Chappaquiddick	Four Poster (7)	25	24	2 (8.33%)	2 (8.33%)
Martha’s Vineyard, Chappaquiddick	Waque (8)	24	24	0 (0%)	0 (0%)
Martha’s Vineyard, Edgartown	Brine’s Pond (9)	25	22	2 (9.09)	2 (9.09)
Martha’s Vineyard, Edgartown	Cape Pogue (10)	25	25	0 (0%)	0 (0%)
Martha’s Vineyard, Edgartown	Correslis State Park (11)	25	25	0 (0%)	0 (0%)
Martha’s Vineyard, Vinyard haven	Cedar Tree Neck (12)	25	25	0 (0%)	0 (0%)
Martha’s Vineyard, Vinyard haven	Seppiesa Point (13)	25	25	0 (0%)	0 (0%)
Total		819	785	33 (4.20%)	33 (4.20%)

**Table 3 viruses-16-00250-t003:** Comparison of Taqman triplex real-time PCR assay, Wadsworth Center singleplex PCR assay, and xMAP MultiFLEX Vector-borne Panel 2 detection of Powassan virus (POWV) in *Ixodes scapularis*.

	Triplex qPCR	Wadsworth Center qPCR	xMAP MultiFLEX Vector-Borne Panel
Total ticks	819	819	819
*n* failed internal control	34	N/A ^1^	N/A ^1^
POWV-positive ticks	33	33	30
POWV-negative ticks	752	786	789

^1^ There is no internal quality control based on *Ixodes scapularis* RNA built into the Wadsworth Center qPCR assay or xMAP MultiFLEX Vector-borne panel 2 assays.

## Data Availability

All data are found within the main body of the manuscript.

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
