# Peer review of "Active Surveillance of Powassan Virus in Massachusetts Ixodes scapularis Ticks, Comparing Detection Using a New Triplex Real-Time PCR Assay with a Luminex Vector-Borne Panel"

_viruses, 2024, doi:10.3390/v16020250_

Round 1

Reviewer 1 Report

Comments and Suggestions for Authors

Xu and colleagues developed a novel TaqMan assay for the surveillance of the prevalence of Powassan virus in Ixodes ticks. The important innovation compared to the currently used qPCR diagnostic is that the assay can distinguish the two lineages of the virus (the Powassan and Deer tick virus) and that it contains a cleverly designed probe for the tick RNA isolation quality control. They compared the new assay against the standard single pex qPCR assay and a commercial Luminex assay (also detecting PCR-amplified pathogen-specific DNAs). The assay was highly specific to the Powassan virus and at least as sensitive as the standard single-plex assay. The authors analysed the ticks collected in different geographic locations of Massachusetts and demonstrated a significant variation in the local prevalence of the Deer tick virus-bearing ticks ranging from 0 to 10%.

The paper is clearly written and describes an important innovation that can be easily applicable for practical Powassan virus surveillance.

I have only a couple of minor comments:   While the PCR-based assays identified three more ticks as positive (33 vs 30) compared to the Luminex-based assay, without the statistical significance it is impossible to conclude that these assays are indeed more sensitive, the results may only suggest that,  and the wording should be changed accordingly (i.e. lines 19, 185,).

 Line 164. Ct is cycle threshold, not critical threshold

Author Response

Xu and colleagues developed a novel TaqMan assay for the surveillance of the prevalence of Powassan virus in Ixodes ticks. The important innovation compared to the currently used qPCR diagnostic is that the assay can distinguish the two lineages of the virus (the Powassan and Deer tick virus) and that it contains a cleverly designed probe for the tick RNA isolation quality control. They compared the new assay against the standard single pex qPCR assay and a commercial Luminex assay (also detecting PCR-amplified pathogen-specific DNAs). The assay was highly specific to the Powassan virus and at least as sensitive as the standard single-plex assay. The authors analysed the ticks collected in different geographic locations of Massachusetts and demonstrated a significant variation in the local prevalence of the Deer tick virus-bearing ticks ranging from 0 to 10%.

The paper is clearly written and describes an important innovation that can be easily applicable for practical Powassan virus surveillance.

I have only a couple of minor comments:   While the PCR-based assays identified three more ticks as positive (33 vs 30) compared to the Luminex-based assay, without the statistical significance it is impossible to conclude that these assays are indeed more sensitive, the results may only suggest that,  and the wording should be changed accordingly (i.e. lines 19, 185,).

The authors thank the reviewer for raising this point on clarity and have revised the language in the manuscript in each occurrence. 

 Line 164. Ct is cycle threshold, not critical threshold

The authors thank the reviewer for raising this point and have corrected Ct to refer to cycle threshold and not critical threshold. 

Reviewer 2 Report

Comments and Suggestions for Authors

The manuscript is devoted to the current problem of choosing the optimal method for identifying the Powassan virus in field material. The internal control proposed by the authors will avoid false negative results. Main note: The authors believe that detection of the virus in more ticks is an indication of the higher sensitivity of their proposed method. Confirmation is required that this is not a false positive result. It would be good to study these samples using RT-PCR and subsequent Sanger sequencing of the PCR product. Perhaps there is not enough virus in the material to be detected using RT-PCR. In this case, it is worth at least discussing this situation. The sensitivity of the methods could be compared simply by diluting the virus-containing tick suspension. Minor note. It would be good to indicate the GPS coordinates of all collection points.

Author Response

The manuscript is devoted to the current problem of choosing the optimal method for identifying the Powassan virus in field material. The internal control proposed by the authors will avoid false negative results. Main note: The authors believe that detection of the virus in more ticks is an indication of the higher sensitivity of their proposed method. Confirmation is required that this is not a false positive result. It would be good to study these samples using RT-PCR and subsequent Sanger sequencing of the PCR product. 

The authors thank the reviewer for raising this point regarding confirmation of positive results and have amended the discussion section accordingly. 

Perhaps there is not enough virus in the material to be detected using RT-PCR. In this case, it is worth at least discussing this situation. The sensitivity of the methods could be compared simply by diluting the virus-containing tick suspension. Minor note. It would be good to indicate the GPS coordinates of all collection points.

The authors thank the reviewer for raising this point and have added some prose to the discussion section to address how we may compare method sensitivity by diluting viral suspension.